# Studies of Nickel/Samarium-Doped Ceria for Catalytic Partial Oxidation of Methane and Effect of Oxygen Vacancy

**Andrew C. Chien** [1,2,*] **, Nicole J. Ye** [1] **, Chao-Wei Huang** [3] **and I-Hsiang Tseng** [1]

[1] Department of Chemical Engineering, Feng Chia University, Taichung 40724, Taiwan; snoopy0302.0302@gmail.com (N.J.Y.); ihtseng@fcu.edu.tw (I.-H.T.)

[2] Green Energy Development Center, Feng Chia University, Taichung 40724, Taiwan

[3] Department of Chemical and Materials Engineering, National Kaohsiung University of Science and Technology, Kaohsiung 80778, Taiwan; huangcw@nkust.edu.tw

[*] Correspondence: cyinchien@fcu.edu.tw; Tel.: +886-424-517-250 (ext. 3691); Fax: +886-424-510-890

**Abstract:** We investigated the performance of nickel/samarium-doped ceria (Ni/SDC) nanocatalysts on the catalytic partial oxidation of methane (CPOM). Studies of temperature-programmed surface reaction and reduction reveal that catalytic activity is determined by a synergistic effect produced by Ni metals and metal-support interaction. Catalytic activity was more dependent on the Ni content below 600 °C, while there is not much difference for all catalysts at high temperatures. The catalyst exhibiting high activities toward syngas production (i.e., a $CH_4$ conversion >90% at 700 °C) requires a medium Ni-SDC interaction with an Sm/Ce ratio of about 1/9 to 2/8. This is accounted for by optimum oxygen vacancies and adequate ion diffusivity in the SDCs which, as reported, also display the highest ion conductivity for fuel cell applications.

**Keywords:** Ni/SDC; microwave; syngas; metal-support; partial oxidation of methane; fuel cell





## 1. Introduction

Catalytic partial oxidation of methane (CPOM) is an important process to convert natural gas into high value-added products. In industry, chemicals such as methanol are produced using syngas (carbon monoxide and hydrogen, CO and $H_2$) from highly endothermic steam reforming reactions [1,2]. Compared with the steam reforming of methane ($CH_4$), CPOM is a preferred technology to produce $H_2$ or synthesis gas due to its fast kinetics and exothermic nature. Moreover, CPOM produces synthesis gas with a stoichiometry of $H_2$/CO of about two, which can be directly used for methanol or Fischer–Tropsch synthesis [3]. On the other hand, many studies have investigated the CPOM in solid oxide fuel cells (SOFCs) when $CH_4$ is used as a fuel [4]. Direct use of $CH_4$ in fuel cells provides a niche for producing electric power and syngas at the same time [5]. Nevertheless, the development of effective catalysts for the CPOM in fuel cells and the synthesis of chemicals remains challenging [4,6].

Supported nickel (Ni) catalysts are widely applied in CPOM due to their low cost and high effectiveness compared to noble metals [2,7,8]. However, the application of the Ni catalyst to CPOM is limited due to the deactivation of Ni nanoparticles by sintering and coking, i.e., carbon deposition [9,10]. Nickel is also an active component in the state-of-the-art anode of SOFCs, i.e., nickel/yttrium stabilized zirconia (Ni/YSZ). Exposure of the anode to CH4 leads to encapsulation of the Ni site by carbon, with subsequent cell degradation [9,11,12]. In practice, CPOM encounters not only coking on the active site but also a competitive reaction, complete oxidation of $CH_4$ to $CO_2$, which is favored thermodynamically. Several reaction mechanisms of CPOM have been investigated [2,4,13], and two of them are proposed. The first is a direct mechanism where $CH_4$ and $O_2$ react on the catalyst surface to yield syngas directly. The second is a combustion-reforming mechanism where $CH_4$ and $O_2$ proceeds in the complete oxidation first, followed by

reforming of excess $CH_4$ by $CO_2$ or steam to syngas. Moreover, it is generally accepted that a metallic site catalyzes the partial oxidation of methane; by contrast, complete oxidation takes place on metal oxides [14].

As mentioned above, efficient Ni catalysts for CPOM require control of active sites over electronic and structural properties, which depend on the active phase as well as the support. Particle size is found to be critical since Ni nanoparticles are more active at lower temperatures, whereas larger Ni particles seem to be more stable with a lower deactivation rate [15]. The effect of the support on the performance of the Ni-based catalysts is also studied [16]. Alumina ($Al_2O_3$) and doped ceria ($CeO_2$) have received attention due to a higher dispersion of the Ni sites on the alumina surface and the presence of oxygen vacancies in the ceria lattice, respectively [17]. Ceria is known to play a promoting role in transitional metals to improve the activities of small metal clusters [18,19]. Excellent catalytic performance of these metal catalysts is attributed to the mechanism of oxygen transfer from ceria to metal, which is strongly dependent on morphology and size of ceria particles, as well as on metal-support interaction [20–22]. For example, a remarkable lattice relaxation (i.e., an increase in the lattice constant), corresponding to facile oxygen mobility, can be observed in ceria nanoparticles compared with the bulk one. Otherwise, doping with a lower valent cation alters the densities of oxygen vacancies in ceria [23], which may increase oxygen adsorption and storage capacity to facilitate the removal of carbonaceous species once they have been deposited.

Combinations of doped ceria and a transition metal, e.g., Ni, have been proven to increase CPOM activity [19,24]. As commonly occurs in catalysis, the activity and stability of metal catalysts depends not only on size and surface structure but also on the intricate properties of the metal-support interaction. $CeO_2$ nanoparticles are therefore preferred compared to bulk materials, as nanoscale ceria tend to have more contact with the metal to liberate and afford more oxygen vacancies, thus exhibiting high oxidation activity [18,25,26]. Further, concentrations of the dopant in ceria not only affect the localized environment but also determine the number of oxygen vacancies [27]. Nonetheless, relevant reports on the effect of doping ratios in ceria are scarce, and most of them are limited to the discussion of physical properties, such as oxygen ion conductivity, in these materials [28–31]. It is understandable that a rational doping level is vital when employing doped ceria as electrolytes in SOFCs, since the doping level correlates with ion conductivity [32]. Although increasing doping levels gives rise to more vacancies, a high level does not necessarily yield a higher conductivity. Above a critical threshold, ion conductivity would drop off due to the limited diffusion of vacancies.

Recently, we synthesized nanoscale Ni/SDC catalysts via microwave-assisted methods and reported the effect of different methods and preparation parameters on CPOM activity [15]. These catalysts contained Ni and SDC, both in nanoscale, and catalyzed a nearly complete oxidation of methane to syngas at 700 °C. In the present study, we manipulated the contents of Ni loading and Sm/Ce ratios to investigate their effect on catalytic performance. Our results showed that reducibility and Ni-SDC interaction play crucial roles in the catalytic activity, and they are ascribed to changes in Ni particle size and oxygen vacancies. The best CPOM activity can be obtained by optimizing the x value in doped ceria, $M_xCe_{1-x}O_{2-\delta}$, (M: trivalent cation), and Ni loadings. These findings provide information on Ni/SDC catalysts for use in other oxidation reactions and allow for better design of other doped ceria catalysts.

## 2. Results

### 2.1. Effect of Ni Content on POM Activity

Figure 1 presents the conversion of methane versus temperature on the catalyst with various Ni contents. The results show that the co-microwave synthesized $NiO/Sm_{0.1}Ce_{0.9}O_{1.95}$ catalyst exhibits high catalytic activity for the oxidation of methane. All catalysts reached a conversion of ~80% above 600 °C. Among them, the SDC catalyst containing 8–12 wt.% Ni gave a higher activity. Examining the effect of temperature on the oxidation activity of

$CH_4$, the conversion increased with increasing temperature. Nevertheless, the catalytic activity was more dependent on the content of nickel at low temperatures (<600 °C), while there is not much difference for all catalysts at high temperatures.

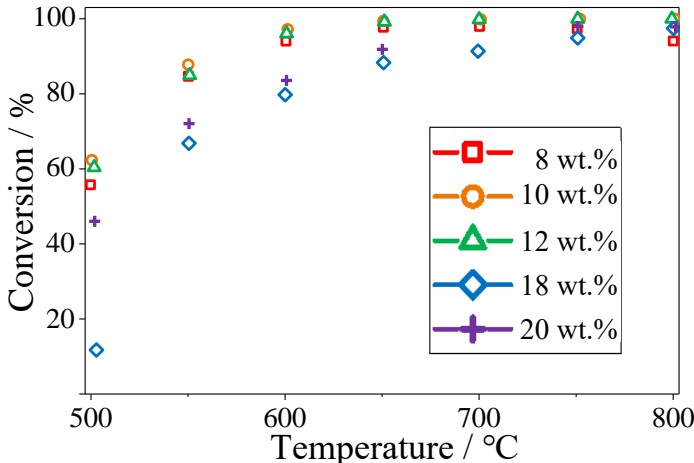

**Figure 1.** The $CH_4$ conversion versus temperature on the $Sm_{0.1}Ce_{0.9}O_{1.95}$ catalysts with various Ni contents in the gas stream, $CH_4/O_2$ (10/5, vol.%), in Ar, at a total flow rate of 30 mL min$^{-1}$.

Figure 2 presents the performance of different catalysts in terms of product selectivity ($H_2$, CO, and $CO_2$) as a function of temperature. The results show that the temperature below 550 °C favored total oxidation of methane on all catalysts, with $CO_2$ as the main product and a $CH_4$ conversion of ~40–60%. As the temperature is elevated up to 600 °C, syngas (CO and $H_2$) was produced, as the catalytic partial oxidation of methane (CPOM) becomes dominant. Comparing the catalysts on CPOM activity and the light-off temperature, the one with 10 wt.% nickel was the most active at 600 °C, exhibiting a conversion of ~90%, as well as a selectivity of 40% and 33% for $H_2$ and CO, respectively (Supplementary Materials Figure S1, shown with product yield). However, the product distribution revealed that the ratio of $H_2$ to CO produced on all catalysts decreased with increasing temperature. The ratio of $H_2/CO$ was close to 1 at 700–800 °C, implying that $H_2$-consuming reactions other than the CPOM reaction were probably involved. The $H_2$ might be reacted by lattice oxygen in these SDC nanocatalysts, which easily adsorb oxygen gas due to their nanoscale nature, since much condensed water was observed in downstream tubings.

Figure 3 shows the temperature-programmed reduction profiles of the SDC catalyst with different nickel contents. The $H_2$-TPR result in the range of 25–500 °C provides information on the supported nickel clusters regarding crystal structure and reduction property because $CeO_2$ is generally characterized by reduction peaks above 500 °C [33]. The TPR profiles show that the catalyst with 8–12 wt.% Ni produces two reduction peaks at low and high temperatures (LT and HT), respectively, while the one with 18–20 wt.% Ni gives only one peak centered at a low temperature ~200 °C. The LT peak was attributed to the reduction of adsorbed oxygen on the SDC and surface NiO sites, whereas the HT resulted from the reduction in internal NiO clusters or the NiO phase interacting with the SDC [7]. The presence of the HT peak was evidence that the reduction in internal NiO sites is a relatively facile process, or that there is a weak interaction between NiO and SDC for a catalyst with low Ni loadings. Specifically, the one with 10 wt. % of Ni displayed the lowest temperature of the HT reduction peak, ~298 °C, implying an easy reduction ability and potentially high activity. On the contrary, the catalyst with high Ni loading tends to cause the agglomeration of nickel species and form large metal crystals (Supplementary Materials Figure S2), which usually take more time and higher temperature for reduction, as shown with a broad $H_2$-TPR profile in Figure 3.

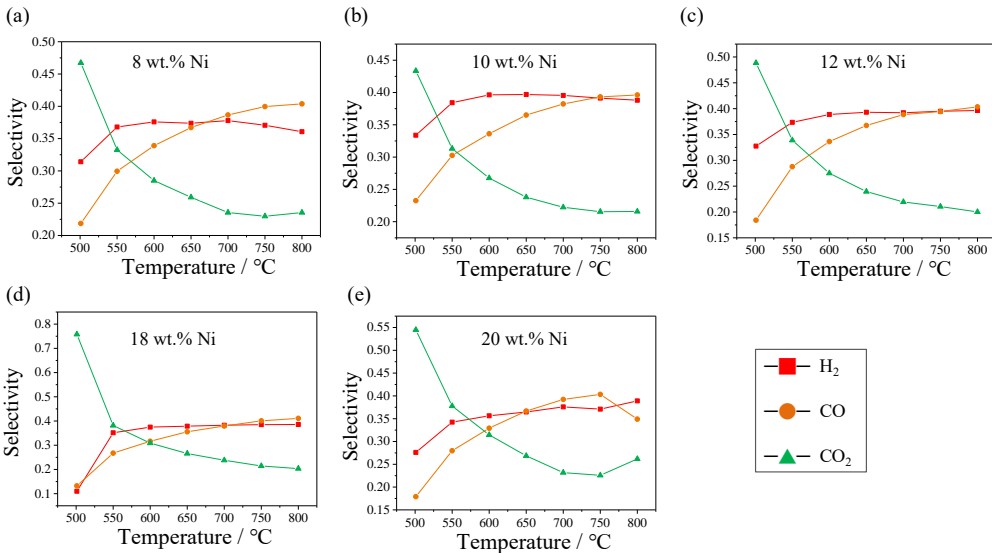

**Figure 2.** The product selectivity versus temperature on the $Sm_{0.1}Ce_{0.9}O_{1.95}$ catalysts with various Ni contents of (**a**) 8 wt.%, (**b**) 10 wt.%, (**c**) 12 wt.%, (**d**) 18 wt.%, and (**e**) 20 wt.% in the gas stream, $CH_4/O_2$ (10/5, vol.%), in Ar, at a total flow rate of 30 mL $min^{-1}$.

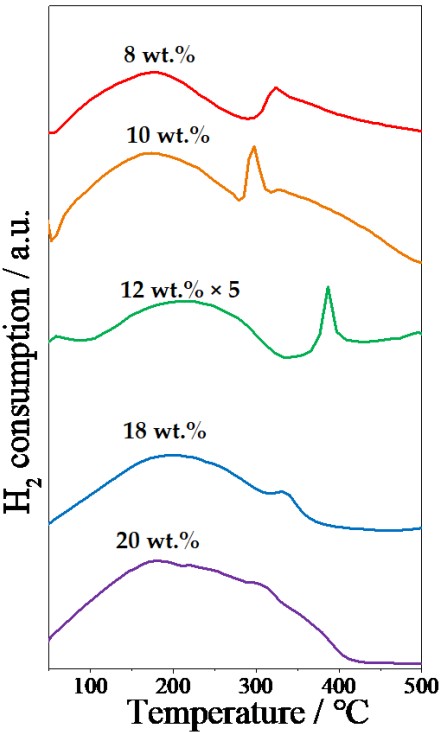

**Figure 3.** The $H_2$-TPR profiles of the $Sm_{0.1}Ce_{0.9}O_{1.95}$ catalysts with different Ni contents (Note: × 5 means original intensity multiplied by 5).

### 2.2. Effect of Sm/Ce Ratio on POM Activity

To investigate effects of Sm doping levels on CPOM activity, the SDC catalysts with different Sm/Ce ratios were synthesized and compared by temperature-programmed surface reaction (TPSR). Figure 4 presents MS profiles of eluting gaseous products from methane oxidation on the Ni/SDC catalyst during the ramping of the temperature to 600 °C. The TPSR result shows that the catalysts with Sm/Ce ratios of 1/9, 2/8, and 3/7 all provoked syngas with a burst production near 500 °C, followed by a progressive drop

in activity. A relatively stable activity of syngas production was then obtained at 600 °C on the catalyst with the Sm/Ce ratios of 1/9 and 2/8, while the one with a ratio of 3/7 was subject to deactivation (note: here, this means relatively stable because of a transient reaction). By contrast, the catalyst of nearly $CeO_2$, i.e., the one with low Sm doping or Sm/Ce = 0.03/0.97, catalyzed the complete oxidation of methane to $CO_2$.

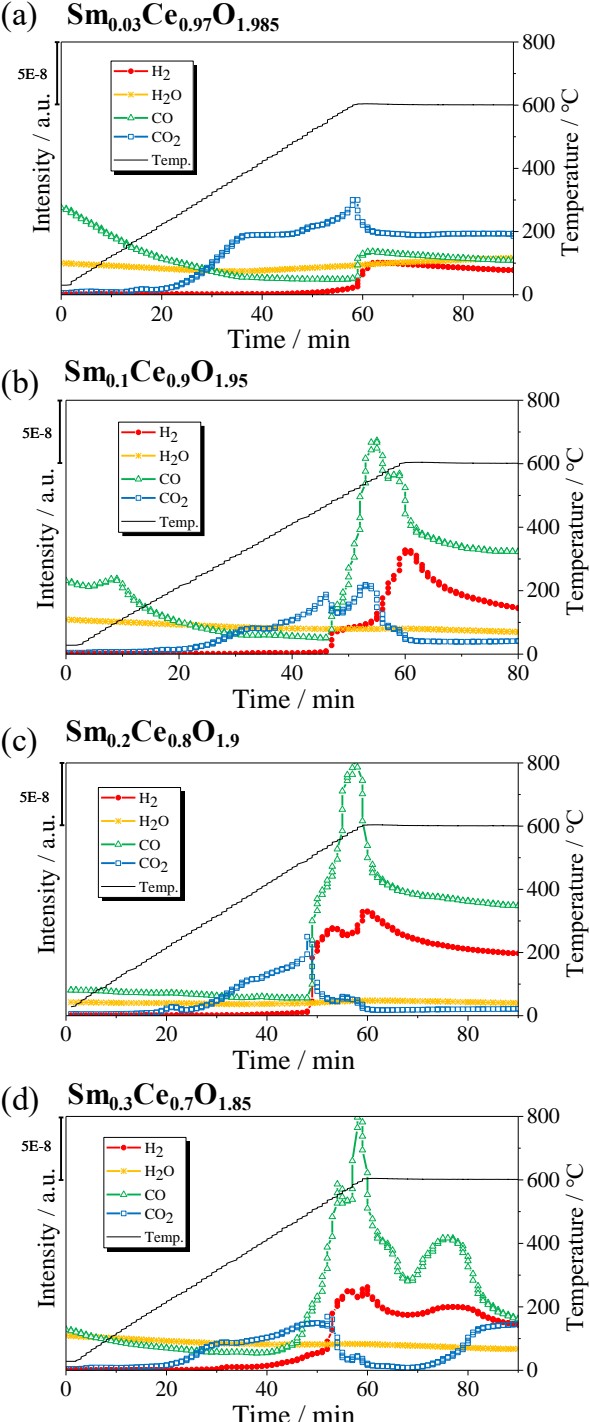

**Figure 4.** MS profiles of eluting gas products from methane oxidation on the co-microwave synthesized 10 wt.% Ni/SDC catalyst prepared with different Sm/Ce ratios. (**a**) 0.03/0.97, (**b**) 1/9, (**c**) 2/8, and (**d**) 3/7.

Figure 5 compares H$_2$-TPR profiles of the Ni/SDC catalyst with different Sm/Ce ratios. The H$_2$-TPR results show that three kinds of reduction zone (labeled as α, β, and γ) can be observed. The low-temperature zone (α) resulted from the reduction in adsorbed oxygen in SDC oxygen vacancies. The α zone merged into zone (β) around 288 °C as the Sm/Ce ratio increased. The γ zone, a right shoulder of β peak extending to high temperature, was attributed to the reduction in surface NiO sites and those weakly interacting with the SDC [7,34]. The shoulder γ peak was shaved down with an increasing Sm/Ce ratio. Higher contents of Sm in the CeO$_2$ raised the reduction temperature of NiO, while the interaction between NiO and SDC is weakened. By contrast, low doping of Sm in the CeO$_2$ decreased the reduction temperature of NiO, whereas the interaction between NiO and SDC remained intense. The catalyst exhibiting optimum interaction between NiO and CeO$_2$ was determined to be the one with an Sm/Ce ratio = 2/8.

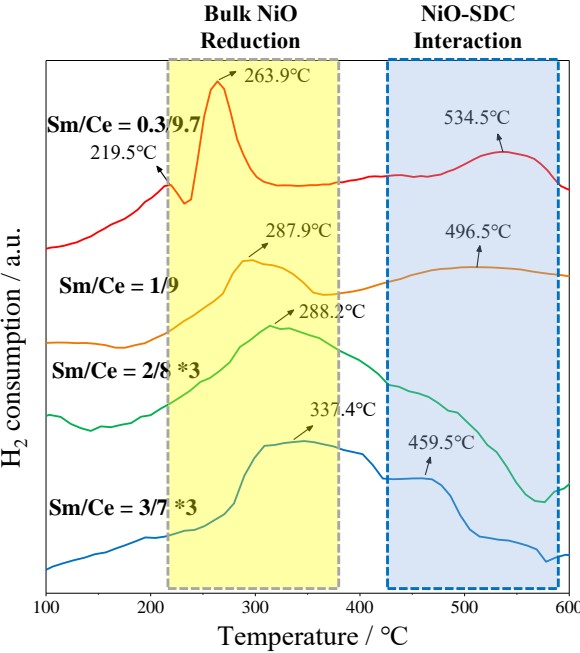

**Figure 5.** The H$_2$-TPR profiles of the co-microwave synthesized 10 wt.% Ni/SDC catalyst prepared with different Sm/Ce ratios (Note: * 3 means original intensity multiplied by 3).

*2.3. Crystalline Structure and Morphology*

Figure 6 examines the indexed powder XRD patterns for Ni/SDC (Sm$_x$Ce$_{1-x}$O$_{2-\delta}$, x = 0.03, 0.1, 0.2, 0.3) nanoparticles. All major Bragg peaks ((111), (200), (220), (311), (222), (400), (331), (420), and (422)) corresponded to the crystallographic plane, having a face-centered cubic fluorite (fcc) structure [27,29]. The peak belonging to a secondary phase, such as Sm$_2$O$_3$, was absent, indicating that there is no impurity with a single-phase formation of doped SDC and the incorporation of Sm$^{3+}$ ions into CeO$_2$ lattice sites (Supplementary Materials Figure S3). As shown in Figure 6, the intensity of the most intense Bragg diffraction peak (111) shifted towards the higher angle side, which is attributable to the lattice contraction for the doping of Sm ions in CeO$_2$. Additionally, the SDC peaks became broader with a decreased crystallinity as the amount of Sm increased. The decreased crystallinity was brought about probably by volume expansion of the CeO$_2$ unit cell due to an increasing level in Sm doping, since the ionic radius of Sm ion (0.108 nm) is larger than that of Ce ion (0.097 nm) [28]. Incorporating Sm into CeO$_2$ resulted in the creation of oxygen vacancies and an increase in lattice parameters relative to pure CeO$_2$, as calculated by Debye Scherrer's formula. On the other hand, the diffraction peaks of NiO were observed only on the sample with Sm/Ce = 1/9 and 2/8. The presence of NiO crystal phase on these samples implied that there likely exists an optimum crystallite NiO size due to interaction

with the SDC. Therefore, these catalysts gave better CPOM activity, as presented in the previous section.

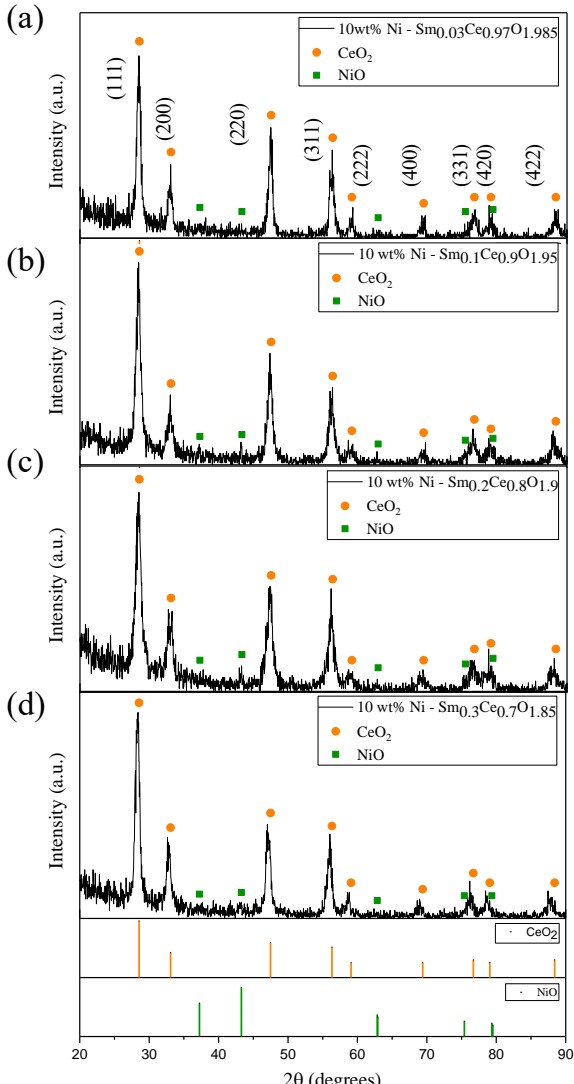

**Figure 6.** The XRD patterns of the synthesized Ni/SDC catalyst with different Sm/Ce ratios, (**a**) 0.03/0.97, (**b**) 1/9, (**c**) 2/8, and (**d**) 3/7.

The crystal particle size and morphology of the as-synthesized samples were analyzed by scanning and transmission electron microscopy (SEM and TEM), as shown in Figure 7. It was clearly seen from the images that the particles on all catalysts were very small crystallites below 20 nm, while some crystallites aggregated together. The particle size became larger with an increasing Sm/Ce ratio, as examined carefully by TEM (shown in the inset of Figure 7). The relationship of particle size versus Sm/Ce ratio agreed well with that of the XRD analysis. Nevertheless, it is challenging to distinguish NiO from SDC, since there is a uniform distribution of particles and crystal phases. The analysis of XRD and EM results both suggested that the NiO species were in nanoscale below 5 nm or partly non-crystalline. These results need to be examined further by high-resolution TEM.

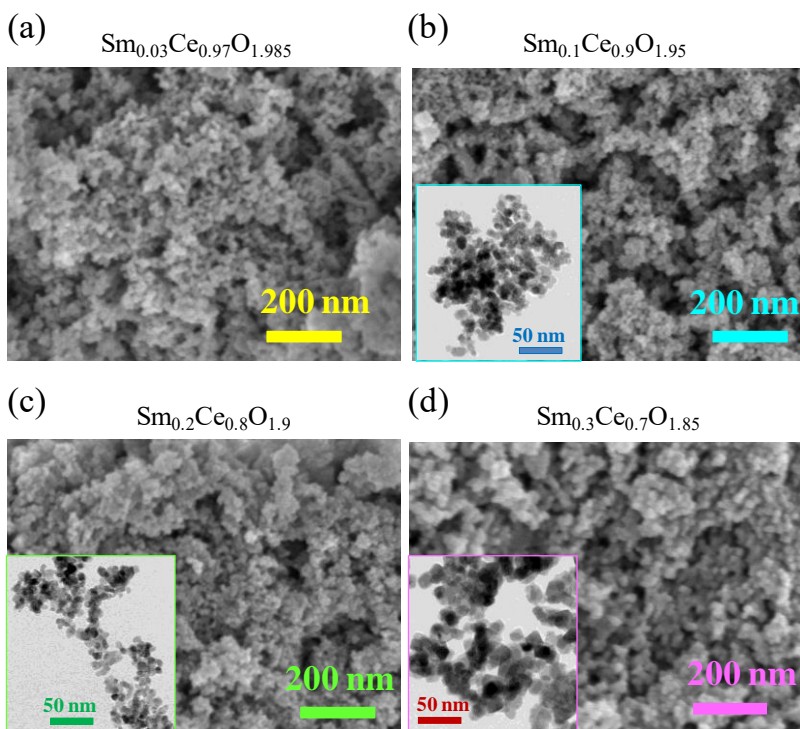

**Figure 7.** SEM images of the as-synthesized Ni/SDC catalyst with different Sm/Ce ratios, (**a**) 0.03/0.97, (**b**) 1/9, (**c**) 2/8, and (**d**) 3/7.

## 3. Discussion

The experimental results above demonstrated that the catalytic activity of Ni/SDC in methane oxidation was greatly affected by the content of NiO and doping levels of Sm in the $CeO_2$. The effect of NiO content on the activity was mainly attributed to the particle size of NiO and its interaction with SDC at low temperatures, i.e., below 600 °C. Increasing the NiO content led to a lower CPOM activity, both for the conversion of $CH_4$ and its selectivity to syngas. The $H_2$-TPR can yield a behavior of bulk reduction, which is usually indicated by the greatest reduction temperature, $T_{max}$, in a TPR profile. As the particle size becomes bigger, the $T_{max}$ shifts to higher temperatures. A large peak also reflects more $H_2$ used for the reduction of a bigger catalyst. The $H_2$-TPR result shows that the catalyst containing high amounts of NiO displays a broad peak extending in the temperature range investigated, indicating that more $H_2$ consumption and a high strength in reduction power are needed. The requirement for high reduction power reflected that an intimate interaction between NiO and SDC or a larger NiO crystallite exists. Large NiO particles usually took more time in the complete reduction of metal clusters underlying the inside. Compared with other catalysts, the 12 wt.% Ni catalyst has a reduction peak at higher temperatures, which was speculated to result from a reduction in surface NiO sites and those weakly interacting with the SDC. Relative to 10 wt.% Ni one, the 12 wt.% Ni catalyst exhibits more surface sites that interact with SDC, thus requiring a higher reduction temperature. The 10–12 wt.% Ni loading on SDC seems to be optimal for POM reaction. As a result, catalytic activity was lower for the catalyst with a higher Ni content at low temperatures, while there is not much difference for all catalysts at high temperatures.

On the other hand, the effect of the Sm/Ce ratio on activity resulted from the density of oxygen vacancies and the interaction between Ni and SDC due to their crystalline characteristics [24]. The catalyst with a suitable doping ratio of Sm/Ce, i.e., 1/9–2/8, produced syngas steadily after the temperature was ramped to 600 °C. The $H_2$-TPR result evidenced that increasing the Sm/Ce ratio weakened the interaction between NiO and SDC, whereas a further increase seemed to trigger complex interactions between these two species. The XRD characterization also demonstrated that the lattice structure was altered;

a trend that is consistent with that observed in the TPR profiles. The doping ratio of Sm/Ce, ~1/9 to 2/8, made a catalyst with an optimum concentration of oxygen vacancies and adequate space for the ion diffusion, which accounts for the high activity toward syngas production. The effect of different Sm doping was also investigated by Raman analysis, as shown in Figure 8. Oxygen vacancies can be evaluated by an absorption at 564 cm$^{-1}$, which is attributed to vibration of Ce–O around Ce center nearby [29,35,36]. We observed no such vibration on pure $CeO_2$ or $Sm_2O_3$ (Supplementary Materials Figure S4). In fact, the SDC with 20 mol.% of Sm doping is also reported to possess the highest ion conductivity in the application of fuel cells [37,38]. Despite its different catalytic activity, the appearance of NiO phase in the XRD pattern was probably also ascribed by the crystalline nature of SDC incurred by different levels of Sm doping. Coincidentally, the NiO phase was observed only on two Ni/SDC catalysts with high activity for syngas production. Whether altering the Sm/Ce ratio brings forth crystallization of NiO in the synthesis environment, meaning that they both produce a synergistic effect responsible for high activity, requires further study.

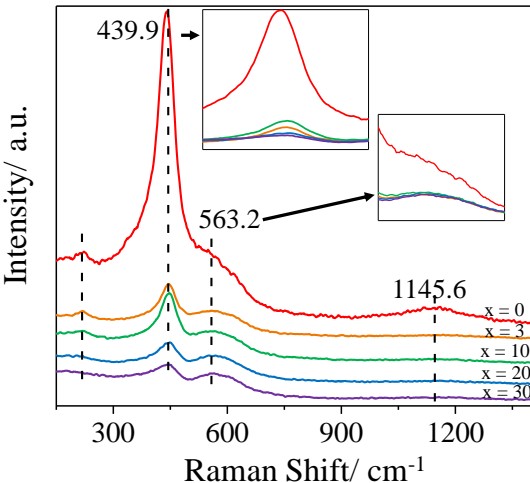

**Figure 8.** Raman Spectra of the as-synthesized 10 wt.% $Ni/Sm_xCe_{1-x}O_{2-\delta}$ catalyst.

## 4. Materials and Methods

### 4.1. Catalyst Preparation

The nickel oxide/samarium-doped cerium oxide (NiO/SDC) was synthesized by a microwave-assisted method [24,39]. In a typical preparation, the precursors of nickel (II) nitrate hexahydrate ($Ni(NO_3)_2 \cdot 6H_2O$), cerium (III) nitrate hexahydrate ($Ce(NO_3)_3 \cdot 6H_2O$), and samarium(III) nitrate hexahydrate ($Sm(NO_3)_3 \cdot 6H_2O$) (All from Sigma Aldrich, St. Louis, MO, USA) were mixed and dissolved together in ethanol. The mixture solution was controlled at pH value ~14 by dropwise addition of sodium hydroxide (NaOH) and placed inside a conventional household microwave. The synthesis started as the mixture was irradiated with microwave strength ~133 W in an on/off cycle (10s on and 20s off) for 10 min. The solid powder precipitated from the irradiated solution was cleaned with distilled water and rinsed with ethanol. The obtained sample was then dried and calcined at 600 °C for 4 h. The product (NiO/SDC) had a yield of about 80 mol.% (+/− 5%) based on cerium. The catalysts containing different Ni contents (8, 10, 18, 20 wt.% based on the NiO/SDC weight) and varying Sm/Ce molar ratios (0.03/0.97, 1/9, 2/8 and 3/7) were prepared and specified. All catalysts were kept dried and ground in the mortar prior to reaction tests and characterizations.

### 4.2. Measurement of Catalytic Activity

The measurement of catalytic activity was conducted in a fixed-bed differential reactor under atmospheric pressure, as described previously [15]. The catalyst of $Ce_{0.1}Sm_{0.9}O_{1.95}$ with different content of nickel was reduced in an $H_2$ environment (2 vol.% in argon gas

flow) at 500 °C for 1 h and then exposed to $CH_4$ (10 vol.%) and $O_2$ (5 vol.%) in Ar (Total flow rate: 30 mL min$^{-1}$). The catalytic activity was studied in $CH_4/O_2$ gas stream by ramping temperature from 500 to 800 °C at a rate of 10 °C min$^{-1}$. The effect of the Sm/Ce ratio on the activity was investigated by a temperature-programmed surface reaction from 25 to 600 °C without further reduction of the catalyst. The effluent gas species were monitored by an online mass spectrometer (Hiden Analytical, HPR 20). Mass/electron (m/e) ratio in the MS were selected for $H_2$ (2), $CH_4$ (15), $H_2O$ (18), CO (28), $O_2$ (32), Ar (39) and $CO_2$ (44). The conversion and selectivity values were taken after 20 min of reaction at each temperature. The $CH_4$ conversion (%) was calculated by $[(C_{CH4,in} - C_{CH4,out})/C_{CH4,in}] \times 100\%$, and the product selectivity was obtained by $[C_{H2,out}/(C_{H2,out} + C_{CO,out} + C_{CO2,out})] \times 100\%$, with $H_2$ as an example.

### 4.3. Characterization

$H_2$ temperature-programmed reduction ($H_2$-TPR) experiments were performed to study reduction behaviors of the catalyst. The TPR experiments were conducted with ~100 mg of the catalyst loaded in a stainless steel tube reactor. The catalyst was first pretreated at 100 °C in a gas mixture with 10 vol.% $H_2$ in Ar (total flow rate: 20 mL min$^{-1}$) and heated at a ramping rate of 5 °C min$^{-1}$ from 25 to 600 °C. Reduction temperature and consumption of $H_2$ was analyzed by the mass spectrometer.

A Bruker D8 Discover diffractometer (Billerica, MA, USA) was used to obtain information of phase composition and crystallinity on the catalyst in the angular range 20–90°, with a scan speed of 2.5° min$^{-1}$. The particle size and surface structure of the as-synthesized NiO/SDC catalyst was examined by a scanning electro n microscope (SEM; JEOL JSM-7800F, Boston, Massachusetts, USA) equipped with an energy-dispersive X-ray spectrometer (EDX, Boston, MA, USA), as well as a transmission electron microscope (TEM; JEM-2010 high resolution). Raman spectra of all the samples were recorded using a Raman imaging microscope (AvaSpec-ULS TEC) with a green laser (532 nm) excitation light source.

### 5. Conclusions

The Ni/SDC catalysts with varying NiO contents and Sm/Ce ratios were studied for catalytic partial oxidation of methane (CPOM). Increasing NiO content from 8 to 20 wt.% decreased the CPOM activity below 600 °C. The catalyst with a high NiO content catalyzed the complete oxidation of $CH_4$ at low temperatures, probably due to the presence of the oxide phase with low reducibility, as evidenced by the $H_2$-TPR results. The insufficient reduction was attributed to larger NiO crystals and their interaction with SDC. Nonetheless, the effects of NiO content became less severe with high reducing power at higher temperatures. On the other hand, doping Sm into ceria led to changes in the lattice parameters and the generation of oxygen vacancies, which provide a means to tailor the activity of ceria catalysts. We have shown that varying Sm/Ce ratios shifts the reaction from a complete to a partial oxidation of methane as the Sm/Ce ratio increases, whereas further increasing the ratio reverts this mechanism. The Sm content in the $CeO_2$ affected CPOM activity by altering the reducibility of NiO on the one hand, and the interaction between NiO and SDC on the other hand. The best CPOM activity has to be optimized by considering these two factors. The information provided in this study allows property control of these microwave-synthesized Ni/SDC nanoparticles, and makes them potential materials in applications such as air control, oxygen storage, and energy.

**Supplementary Materials:** The following are available online at https://www.mdpi.com/article/10.3390/catal11060731/s1, Figure S1: Product yield of on the SDC catalysts with various Ni contents, Figure S2: XRD pattern on (a) 10 wt. % Ni/SDC and (b) 15 wt.% Ni/SDC catalyst, Figure S3: Experimental Ce3d and Sm 3d XPS spectra on 10 wt.% Ni/SDC (Sm/Ce=1/9), Figure S4: Raman analysis on the commercial CeO2 and as-synthesized SDC.

**Author Contributions:** Conceptualization, A.C.C., N.J.Y.; methodology, A.C.C., N.J.Y.; software, N.J.Y.; validation, formal analysis and investigation, A.C.C., N.J.Y.; writing—original draft preparation, A.C.C.; writing—review and editing, C.-W.H., I.-H.T. All authors have read and agreed to the published version of the manuscript.

**Funding:** This research was funded by Ministry of Science and Technology in Taiwan under contract MOST 109-2221-E-035-021-MY3 and MOST 109-2622-E-035-019.

**Data Availability Statement:** Data is contained within the article or supplementary material.

**Acknowledgments:** The authors would like to acknowledge the administrative support from Feng Chia University.

**Conflicts of Interest:** The authors declare no conflict of any personal circumstances or interest that may be perceived as inappropriately influencing the representation or interpretation of reported research results. The funders had no role in the design of the study; in the collection, analyses, or interpretation of data; in the writing of the manuscript, or in the decision to publish the results.

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
