# Peer review of "Studies of Nickel/Samarium-Doped Ceria for Catalytic Partial Oxidation of Methane and Effect of Oxygen Vacancy"

_catalysts, doi:10.3390/catal11060731_

Round 1

Reviewer 1 Report

The authors report on the catalysts for partial methane oxidation with the nickel as the active phase supported on a samarium-doped ceria. The subject, in general, is interesting, however the presented manuscript has certain deficiencies.

  • Lack of the proper literature overview with recent results, with respect to the catalytic partial oxidation of methane.
  •  Figure 1 and presented therein CH4 conversion is rather irrelevant, the conversion to CO/H2 is important. Moreover, similar reactivity determination (conversion to desired products) is missing for the samples with fixed Ni content and different Sm doping level.
  • The authors mention other reactions may take place (l.107) - what other reactions?
  • Figure 2 does not allow to directly compare the catalysts (see also above).
  • How can H2-TPR provide information about crystal structure?
  • Figure 3 - why does 12 wt% Ni sample stand out in the series?
  • The authors write that stable activity is achieved at 600oC (l.141) while a decrease is observed. The long-term stability appears to be an issue here.
  • Why weakened interaction of NiO with the Sm/CeO2 should result in a higher temperature of NiO reduction? How do the authors determine strength of the interaction of the NiO and SDC (l.155)? How was the optimum interaction between NiO and Sm/CeO2 determined?
  • How do the authors define crystallinity (l.172)?
  • The quality of XRD patterns is low - it is difficult to determine whether NiO is present or not.
  • There is lack of quantitative analysis of any kind. What about  the H2-TPR quantitative data analysis? Quantitative analysis should be added.
  • "The effect of NiO content on the activity was mainly attributed to the particle size of NiO" (l.199) There is no data presented to support this conclusion.
  • What do the authors mean by "intimate interaction between NiO and SDC or a larger NiO crystallite"? (l.205) In general, in many places of the manuscript the interaction of NiO with the support is mentioned, but its nature is unclear.
  • Why longer reduction time results in decreased activity? (l.206) Catalysts are active in a reduced state, so aren't they completely pre-reduced prior the reaction? If not - why?
  • The concentration/density of oxygen vacancies were not investigated/determined, thus they should not form the basis for any conclusions.
  • Why do the authors think high ionic conductivity of the support is important for CPOM?

Author Response

The response letter is attached.

Reviewer 2 Report

Reviewer’s Report

Studies of Nickel/Samarium Doped Ceria for Catalytic Partial Oxidation of Methane and Effect of Oxygen Vacancy

Manuscript ID: catalysts-1230888

In this manuscript, the authors investigated the performance of nickel/samarium doped ceria (Ni/SDC) nanocatalysts on catalytic partial oxidation of methane (CPOM). The results presented in the manuscript are interesting and worth publishing. Therefore, I recommend the manuscript for publication with the following amendments.

  • Provide the crystallite size and lattice parameters of the catalysts.
  • It is necessary to provide the Raman and/or O2-TPD experiment/s to characterize the oxygen vacancies in the samples.
  • Provide more details for characterization experiments. Move the H2-TPR experiment details to the characterization section.
  • Some good papers on ceria-based catalysts are missing.

Chem. Rev. 2016, 116, 5987−6041; Catalysis Reviews 2018, 60, 177-277; ChemSusChem 2010, 3, 654 – 678

  • Some typos could be found. The authors should correct those errors before publication.

Author Response

The response letter is attached.

Reviewer 3 Report

In the presented manuscript, some statements are lacking experimental support, no matter how sound or likely they might seem. There are structural observations shown for the as prepared samples, bit it was not repeated for the spent catalysts. In this context, a chemical analysis is strongly missing here as well.   
In the Introduction the important issue of catalyst stability is mentioned, but in the following it is not addressed at all (which corresponds with the above objection).
It is otherwise well-structured and decently written manuscript, but at some points it is too speculative due to the missing analyses. It has a high potential to be published in the Catalysts journal, but I would strongly recommend authors to provide complementary experimental data as suggested. As such, it makes an impression of a half-done work.   

Introduction:
Lines 55-57: "Alumina (Al2O3) and doped ceria (CeO2) have received attention due to higher dispersion of the Ni sites on the alumina surface and presence of oxygen vacancies in the ceria lattice, respectively" - It is not explained, however, why the presence of oxygen vacancies is considered beneficial. Isn't it rather the high redox activity of ceria, or, if you prefer, facile oxygen vacancy formation on ceria? Similarly on l. 69 ("... determine the number of oxygen vacancies.")  

Results and discussion:
- l. 89 - What is the point of the first sentence? ("The text continues here.")
- l. 128-129 - "the catalyst with a high Ni loading tends to cause the agglomeration of nickel species and form large metal crystals,...." - How do the authors know? No relevant data are provided to support this claim.
- l. 168-169 - "indicating that there is no impurity with single phase formation of doped SDC and the incorporation of Sm3+ ions into CeO2 lattice site." - It is a bit confusing. If the authors were trying to say that the Sm3+ cations are embedded within ceria and not existing with a separate oxide phase, than the sentence needs to be reformulated.
- Moreover, when speaking of particular valence states, it should be supported by a relevant experimental evidence (e.g. by XPS). Otherwise it is a mere guessing.
- A chemical state analysis is also missing to support all the statements mentioning oxygen vacancy formation or concentration changes, as well as the discussions involving metal-support interactions.
- Related to the above, why not to do the actual measurement instead of saying "which need further examined by high-resolution TEM"? This makes no sense to me. We the authors rushed for some reason?    
- l. 215 - "... further increase seems to trigger complex interactions of these two species" is a meaningless statement with no support in the experimental observations. Once again, this is a result of the missing complementary information.

Materials and Methods:

- l. 253-254 - "Mass/electron (m/e) ratio in the 253 MS were selected for H2 (2), He (4), CH4 (15), H2O (18), CO (28), O2 (32), and CO2 (44)." - How did authors account for the molecular cracking and the overlaps in the mass spectra? The mass-molecule assignmets are not always as straightforward as indicated. 

Other minor issues:
- Fig. 1 - Consider adding a data interpolation curves for better clarity.
- Fig. 2 - Modify the legends to just refer to a molecular formulae and move the Ni content to either caption or to the labels of each plot. 
- l. 161 - "profiles of the on the co-microwave synthesized"
- Fig. 3 - What is the meaining of the "* 5" in one of the labels? The same in Fig. 5, bottom 2 curves.
- In some molecular formulae the numbers are not in a subscript.

Language and style:
- The language quality of the manuscript is not bad, yet some improvements could be made. For instance, there are several wrong used plurals, missing or incorrect articles, wrong tenses, wrong prepositions etc.

Author Response

The response letter is attached. Many thanks for the reviewer. Very appreciate!

Round 2

Reviewer 1 Report

Dear Authors,

Apart from the introduction of updated references and some cosmetic changes, you have not made the manuscrscit clearer nor improved the quality of the data. Therefore I am bound to recommend rejection of your manuscript.

Author Response

See attached file and revised manuscript.

Reviewer 2 Report

Accept it in the present format.

Author Response

Thanks for reviewer's consideration.

Reviewer 3 Report

After I had evaluated the manuscript by Chien et al. as a "half-done work" lacking complementary analysis and, at some points, proper discussion, the authors came with a revision which is nearly untouched from the original submission - The only apparent changes in the text are on lines 287-291, plus a minor reference update and a slight visual improvement of the Fig. 2. To me this is largely ignoring the reviewers' comments (not only mine), no matter how they were answered in the reply to the comments, almost nothing has changed from the reader's perspective. If there are some other changes not marked in the revised version, I'm not going to search for them. By the way, a template repetition of the "Thanks for reviewer’s valuable comment." sentence prior to each reply doesn't make a good impression either.
Yet, some of the author's responses are not fully satisfactory, such as to the comment #2 (where issues regarding oxygen vacancies, oxygen mobility, oxide NP vs. bulk are mixed without a rational explanation). In the comment #6, the authors state: "We do have XPS analysis (data was not shown in the manuscript). These data were combined with other analysis and prepared for another manuscript". Why not to show the relevant data in this manuscript, especially if they have been asked for by a reviewer? Saving them for another article smells like a LPU (least publishable unit) practice. The same applies to the comment #7 - you might have a potential supporting data but you decide to hide them from the reader (and rather write a plain unsupported statement in the text, hoping the reader will believe it)?
As written in the reply to the comment #9: "We agree with the reviewer that more direct evidence is needed. As the XPS data shown above, further characterization and experiments are designed to unravel interactions between metal support." I also agree, they are needed, so once you gave them and you are willing to provide them public, go ahead and submit a complete manuscript. In its present form I do NOT recommend the presented manuscript for publication.       

Author Response

see attached file and revised manuscript.

Round 3

Reviewer 1 Report

I can see the additional characterization data now. However, Figure 1 and 2 should still be modified to enable direct comparison of the catalysts' reactivity.

Author Response

Dear Editor,

I would like to thank you and reviewers for the valuable inputs.  We calculate the conversion of products, H2、CO、CO2 on all catalysts, except Sm/Ce catalysts. They are provided in supporting information, Fig. S1. For those Sm/Ce ones, indeed it’s difficult to compare since the reaction is transient.

Sincerely,

Andrew C. Chien

Assistant Professor
Department of Chemical Engineering
Feng Chia University

Reviewer 3 Report

Finally the Supporting information has been added making the manuscript more complete. Now I can also see the modified parts of the document. In its present form it can be re-considered for publication.
I have found just a few other minor issues which should be addressed:
- Provide the fitting of the Ce 3d XP spectra (Fig. S2a). The information value of a complex spectra such as Ce 3d is not very high without spectral deconvolution, from which a abundance of Ce3+ and Ce4+ oxidation states can be estimated. Clearly the spectrum in Fig. S2a shows comaprable amounts of both but more accurate conclusion is impossible to make just from the raw data. Moreover, the u0 and v0 are not indicated in the figure.
- Fix the typo "poewr" in Fig. S2a
- I'm missing references to important papers dealing with ceria redox properties in the noble-metal-ceria systems and the oxygen spillover phenomena on such systems. Especially considering the contributions of the research groups of profs. Libuda, Neyman, Matolin, Fabris, Pirovano and others, their research shouldn't be omitted in the context of the presented work. Particularly relevant for the manuscript topic might be, e.g.,    
Nature materials 10 (4), 310-315, J. Chem. Phys. 151 (20) (2019) art. # 204703, Chemical Communications 46 (32), 5936-5938.

Author Response

I would like to thank you for the valuable inputs.  We have revised the manuscript from the reviewers’ comment accordingly (with revised manuscript). 

Thanks for reviewer’s re-consideration. We fix the typo and label u0 and v0 in XPS. For the references mentioned above, they are valuable and all included.

Sincerely,

Andrew C. Chien

Assistant Professor
Department of Chemical Engineering,

Feng Chia University